# Comparative Study of Physicochemical Properties of Finely Dispersed Powders and Ceramics in the Systems CeO$_2$–Sm$_2$O$_3$ and CeO$_2$–Nd$_2$O$_3$ as Electrolyte Materials for Medium Temperature Fuel Cells

**Marina V. Kalinina [1], Daria A. Dyuskina [1], Sergey V. Mjakin [2,3,\*](image), Irina Yu. Kruchinina [1,4] and Olga A. Shilova [1,4]**

[1]  Institute of Silicate Chemistry of the Russian Academy of Sciences, 2 Makarova Emb, 199034 St. Petersburg, Russia; tikhonov_p-a@mail.ru (M.V.K.)

[2]  Department of Theory of Materials Sciences, Saint-Petersburg State Institute of Technology, Technical University, 24-26/49A Moskovsky Prospect, 190013 St. Petersburg, Russia

[3]  Institute for Analytical Instrumentation, Russian Academy of Sciences, 31-33A Ivana Chernykh Str., 198095 St. Petersburg, Russia

[4]  Department of Nanotechnology and Nanomaterials for Radioelectronics, Saint-Petersburg State Electrotechnical University "LETI", 5 Professora Popova Str., 197376 St. Petersburg, Russia

**\***  Correspondence: svmjakin@technolog.edu.ru

**Abstract:** Finely dispersed $(CeO_2)_{1-x}(Sm_2O_3)_x$ (x = 0.05, 0.10, 0.20) and $(CeO_2)_{1-x}(Nd_2O_3)_x$ (x = 0.05, 0.10, 0.15, 0.20, 0.25) powders were synthesized via liquid-phase techniques based on the co-precipitation of hydroxides and were used to obtain ceramic materials comprising fluorite-like solid solutions with CSR in the range 69–88 nm (upon annealing at 1300 °C) and open porosity in the range 0.6–6.2%. The physicochemical properties of the synthesized materials were comparatively characterized. In general, the prepared materials were found to possess a mixed type of electrical conductivity, but in the medium-temperature range, the ionic component was predominant (ion transfer numbers $t_i$ = 0.93–0.73 at 300–700 °C). The highest ionic conductivity was observed for CeO$_2$-based samples containing 20 mol.% Sm$_2$O$_3$ ($\sigma_{700°C}$ = 3.3 × 10$^{-2}$ S/cm) and 15 mol.% Nd$_2$O$_3$ ($\sigma_{700°C}$ = 0.48 × 10$^{-2}$ S/cm) was in the temperature range 500–700 °C. The physicochemical properties (density, open porosity, type and mechanism of electrical conductivity) of the obtained ceramic materials make them promising as solid oxide electrolytes for medium temperature fuel cells.

**Keywords:** co-precipitation of hydroxides; oxides; finely dispersed powders; nanoceramics; density; porosity; electric properties; fuel cells; electrolytes

## 1. Introduction

A permanently growing demand for power sources has contributed to the deterioration of the worldwide environmental situation. This problem can be addressed by the development of effective and environmentally friendly power generation technologies involving the safe application and disposal of power sources and the products of their conversion. A particularly promising approach was based on the implementation of highly efficient and inexpensive solid oxide fuel cells (SOFC), which afforded an effective direct conversion of the chemical energy of organic fuels into electric power. SOFCs with power from 1 W to 1 kW were particularly important for various mobile and portative devices (electric cars, gadgets, etc.), especially in remote areas lacking centralized power supply systems.

The direct electrochemical conversion of the fuel was promising with respect to their environmental protection since these processes feature a minimal yield of harmful compounds in the absence of intensive noise and vibrations compared with internal combustion

engines and other types of power-generating systems [1–3]. Electrochemical fuel cells convert hydrocarbon fuels into power without any yield of heavy particles, nitrogen oxides, sulfur compounds and other contaminants contributing to the formation of smog and acid rains [4,5], thus providing a "clean" power generation. In view of these factors, an essential goal is the development of components for medium-temperature fuel cells, which can be useful for power generation using all types of hydrocarbons when converted into the synthesis gas ($H_2$-CO). Particularly, one of the main components of fuel cells is electrolytes. According to the ion transport mechanisms, they can be divided into anionic, proton and ion-mixed ones. The operation of medium- and high-temperature fuel cells is based on the transport of oxygen ions ($O_2^-$) from the cathode to the anode. This process is only possible in the presence of oxygen vacancies and requires the application of electrolytes containing anionic vacancies in the crystal lattice [6–8].

However, in order to impart solid oxide electrolytes with an electrical conductivity level that is acceptable for SOFC operation, they should be heated to very high temperatures. For example, for electrolytes based on yttrium-doped zirconia, these temperatures are about $900-1000\ °C$. In recent decades ceria-based electrolytes featuring quite a high conductivity in the medium temperature range ($500-700\ °C$) were considered as an alternative to zirconia-based systems. The use of $CeO_2$-based electrolytes can provide both a decrease in the SOFC working temperature by 300–400 °C and enhance their efficiency.

Particularly promising for SOFC are electrolyte materials that possess high overall electric conductivity in couple with an optimal ionic conductivity, such as ceramics in the systems $CeO_2$–$Sm_2O_3$ and $CeO_2$–$Nd_2O_3$. Hence, the development of novel ceria-based electrolytes with the required conductivity values is an important goal in materials science. Generally, the preparation of solid electrolytes with optimal exploration performances (ionic conductivity, gas density, thermal stability, mechanical strength) requires the application of finely dispersed powders [9–13]. The electrical properties of the considered electrolytes depend on numerous factors, including the applied synthetic procedure, dispersion of the precursor powders, ceramic material density, grain size, etc. [14–18].

The most inexpensive and simple approaches to obtaining nanopowders include liquid phase methods such as hydrothermal synthesis [19], the sol-gel method [20], the co-precipitation of hydroxides from the solutions of inorganic salts and co-crystallization of salts [21–23]. The co-precipitation of hydroxides coupled with low-temperature treatment is the most promising procedure since it provides the most precise control over the dispersion and microstructure of the target products and can obtain weakly agglomerated xerogels and nanoscale powders with high specific surface area [10,21,22].

An available approach to reduce the sintering temperature, increase the density and mechanical strength as well as reduce the porosity of ceramics is based on the controllable addition of sintering additives to the charge mixture. Such additives can play an important role even at a very small content according to different mechanisms and depending on the nature of the base material and the additive, as well as regarding their high-temperature interaction. Additives can activate with this sintering process by blocking grain the growth, being localized at grain boundaries in an initial state or in the form of a compound [21].

Although this problem has been addressed in a number of works, no comparative data have been reported on the relationships among the synthesis conditions, microstructure and electromigration properties of solid electrolytes based on $CeO_2$–$Sm_2O_3$ and $CeO_2$–$Nd_2O_3$ systems.

The aim of this research is the synthesis of finely dispersed $(CeO_2)_{1-x}(Sm_2O_3)_x$ (x = 0.05; 0.10; 0.20) and $(CeO_2)_{1-x}(Nd_2O_3)_x$ (x = 0.05; 0.10; 0.15; 0.20, 0.25) powders by the liquid phase method based on the co-precipitation of the corresponding hydroxides, followed by low-temperature treatment and a comparative analysis of the effect of the elemental and concentration composition on the microstructure and physicochemical properties of the powders and ceramics based thereon.

## 2. Materials and Methods

### 2.1. Synthesis of $(CeO_2)_{1-x}(Sm_2O_3)_x$ (x = 0.05; 0.10; 0.20) and $(CeO_2)_{1-x}(Nd_2O_3)_x$ (x = 0.05; 0.10; 0.15; 0.20; 0.25) Powders by Co-Precipitation of Hydroxides

The liquid phase synthesis of the xerogels and nano-dispersed $(CeO_2)_{1-x}(Sm_2O_3)_x$ (x = 0.05; 0.10 and 0.20) and $(CeO_2)_{1-x}(Nd_2O_3)x$ (x = 0.05; 0.10; 0.15; 0.20; 0.25) powders was performed by the co-precipitation of hydroxides with low-temperature processing.

For this synthesis, the nitric acid salts of cerium $Ce(NO_3)_3·6H_2O$ (analytical purity grade with the reagent content higher than 98% wt.), samarium $Sm(NO_3)_3·nH_2O$ (analytical purity grade with the reagent content higher than 98% wt.) and neodymium $Nd(NO_3)_3·6H_2O$ (chemical purity grade with the reagent content higher than 99% wt.) were used, from which the diluted (~0.1 M) solutions were prepared.

The co-precipitation of hydroxides was carried out according to the scheme shown in Figure 1 using a 1 M aqueous solution of ammonia hydrate ($NH_3·H_2O$) as a precipitating agent.

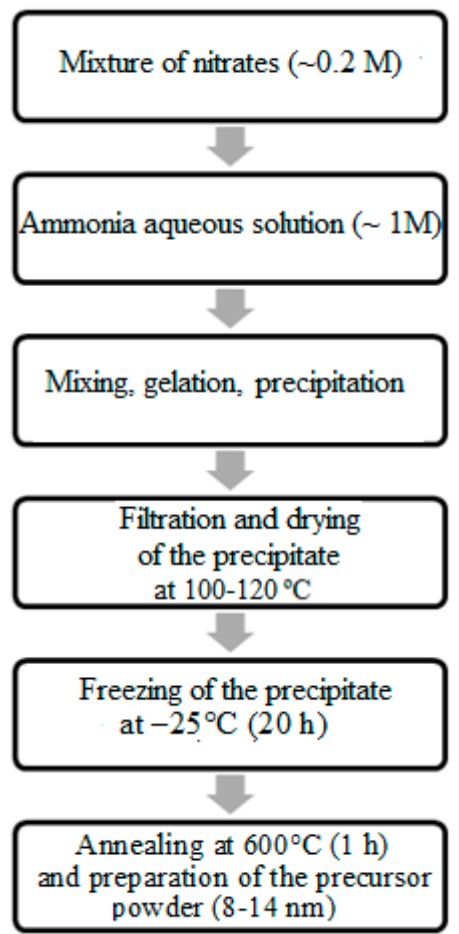

**Figure 1.** Synthetic approach to obtaining $CeO_2$–$Sm_2O_3$ and $CeO_2$–$Nd_2O_3$ ceramic materials.

Taking into account the pH values required for the precipitation of each hydroxide, the pH of the reaction mixtures for Ce-Sm and Ce-Nd systems was maintained on levels 10–11 and 11–12, respectively. This precipitation was performed with a minimal rate of 0.02 cm$^3$/s and thorough mixing. The prepared gelatinous precipitate of hydroxides was filtered, followed by freezing at −25 °C within 24 h to provide deagglomeration and maintain a fine dispersion of the prepared co-precipitates. The freezing of these gels provided the removal of the adsorbed and crystallization water from the gelatinous precipitate, as well as its fastest hardening to maintain a high chemical homogeneity of the solid phase. The application of low-temperature processing determined the evolution

of the product's microstructure and allowed for the preparation of more finely dispersed materials [13].

The resulting X-ray amorphous xerogels were dried at 120 °C within 1 h, followed by heating at 600 °C within 1 h to form nanopowders with a stable crystal structure. The prepared powders were consolidated by single-axis cold compression at the pressure of 150 MPa followed by sintering at 1300 °C within 2 h.

### 2.2. Characterization Methods

XRD analysis was performed using a D8-Advanse diffractometer (Bruker, Billerica, MA, USA), WINFIT 1.2.1 software for data processing with the Fourier transformation and an international database ICDD-2006 for the diffraction patterns interpretation. The size of coherent scattering regions (CSR) was estimated using the Selyakov–Scherrer equation:

$$D_{CSR} = 0.9 \cdot \lambda / (\beta \cdot \cos\theta)$$

where $\lambda$ is the CuK$\alpha$ wavelength and $\beta$ is the diffraction peak FWHM [24].

The thermolysis processes in co-precipitated xerogels and powders were studied at heating in the temperature range of 20–1000 °C using a Q-1000D MOM derivatograph (Paulik-Paulik-Erdey, Budapest, Hungary). The specific surface area of the synthesized nanopowders was characterized according to the BET model by low-temperature nitrogen adsorption using a QuantaChrome Nova 4200B analyzer. The pore size distribution was characterized based on the obtained nitrogen desorption isotherms according to the Barrett–Joyner–Khalenda (BJK) method. Heat treatment of these samples was carried out using a Naberterm oven with programmed heating in the range of 25–1300 °C within 1–2 h, followed by slow cooling in an oven.

The open porosity and apparent density were measured by hydrostatic weighing in distilled water according to the Russian standard GOST 473.4-81 [25].

The surface functionality of the prepared samples was studied using a dynamic pH-metry technique [26,27]. The acid-base properties of the powder surface were characterized by measuring the changes in the pH value of the suspensions obtained by immersing 30 mg of the studied powders in 30 mL of distilled water at permanent agitation with a magnetic stirrer. pH measurements were performed using Multitest IPL-301 pH-meter (NPP SEMICO, Novosibirsk, Russia) in 5, 10, 20, 30, 40, 50 and 60 s after the sample immersion and subsequently, every 30 s up to 5 min from the powder immersion.

The electrical resistance of the obtained ceramic materials was measured by a two-contact method using a direct current in the temperature range of 250–1000 °C and using the "Hardware-software installation to investigate the electrical properties of nanoceramics in different gas media" [28]. The transfer numbers of the ions and electrons in bulky solid electrolytes were determined by the West–Tallan method [29] using a $CO_2$ + CO mixture (corresponding to the oxygen partial pressure of $10^3$ Pa) as an inert gas. The measurements were carried out using a direct current in weak (U = 0.5 V) fields after a long (up to 30 min) drop of the current. The contributions of ionic and electronic conductivity were estimated as:

$$t_e = R_{air} / R_e$$

and

$$t_i = 1 - t_e$$

where $t_e$ and $t_i$ are the transport numbers of the electrons and ions, respectively, and $R_{air}$ and $R_e$ are the sample resistance measured in air and in an inert gas atmosphere.

The resulting ceramics microstructure was characterized using a Tescan Amber GMH (Tescan, Chech Republic) electron microscope with a secondary electron detector (Everhard-Tornley) at magnification $\times 75,000$.

## 3. Results and Discussion

### 3.1. Study of the Thermolysis of the Synthesized Xerogels

The thermal behavior of the synthesized xerogels was studied using differential thermal analysis up to 1000 °C (heating rate 20 °C/min). An exemplary DTA thermogram is shown in Figure 2 for a $(CeO_2)_{0.95}(Sm_2O_3)_{0.05}$ xerogel obtained by co-precipitation without (Figure 2a) and with subsequent freezing (2b). The endothermic effect with a minimum at ~110 °C in Figure 2a is due to the removal of solvent residues and the desorption of physically bound water from the surface of xerogel particles. The dehydration of crystalline hydrate and decomposition of nitrate salts occurred in one stage, corresponding to a weight loss of ~ 33%. In the temperature range of 260–280 °C, these powders exhibited a narrow exothermic effect due to the crystallization of a fluorite-like cubic solid solution based on cerium oxide, which was accompanied by a small weight loss, probably associated with the removal of water through the pores that were formed as a result of the transformation of the xerogel microstructure. However, for the sample subjected to freezing, no endothermic effect corresponding to a similar dehydration process was observed (Figure 2b), suggesting that freezing led to the removal of most of the water. Freezing also provided a decrease in the temperature range of ceria-based solid solution crystallization from 260–320 °C to 230–280 °C. The weight loss of the crystalline hydrate prepared without freezing (Figure 2a) was ~11%.

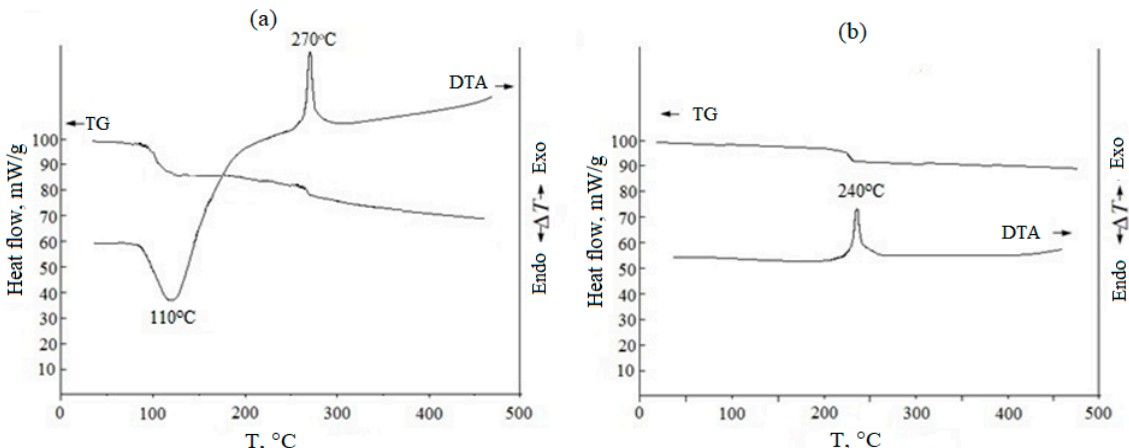

**Figure 2.** TG/DTA results for $(CeO_2)_{0.95}(Sm_2O_3)_{0.05}$ xerogels prepared without (**a**) and with (**b**) freezing at −25 °C within 24 h.

An exemplary DTA thermogram for $(CeO_2)_{0.80}(Nd_2O_3)_{0.20}$, prepared by co-precipitation, followed by freezing, is shown in Figure 3.

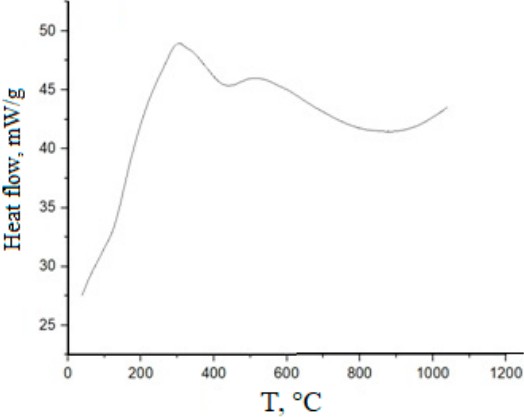

**Figure 3.** DSC plot for $(CeO_2)_{0.80}(Nd_2O_3)_{0.20}$ powder prepared by co-precipitation, followed by freezing.

In this case, an exothermic effect corresponding to dehydration at ~100 °C was not observed, suggesting the removal of physically sorbed water in the course of freezing. Peaks relating to the desorption of structurally bound water and the decomposition of residual nitrates are present in the range 320–390 °C. A narrow exothermic peak at 260–280 °C could be determined by the crystallization of the cubic solid solution.

### 3.2. Characterization of the $CeO_2$–$Sm_2O_3$ and $CeO_2$–$Nd_2O_3$ Powders Microstructure

An exemplary adsorption–desorption isotherm and differential pore size distribution for a $(CeO_2)_{0.95}(Sm_2O_3)_{0.05}$ precursor powder dried at 120 °C is presented in Figure 4. As can be seen from Figure 4a, the powder had a mesoporous structure, as evidenced by the adsorption–desorption isotherm relating to type IV according to the IUPAC classification. The type of capillary-condensation hysteresis for the H2 type, according to the IUPAC classification, indicated the predominance of bottle-shaped pores, mainly small mesopores (2–10 nm). The total pore volume was 0.083 cm$^3$/g, and the specific surface area was 50 m$^2$/g.

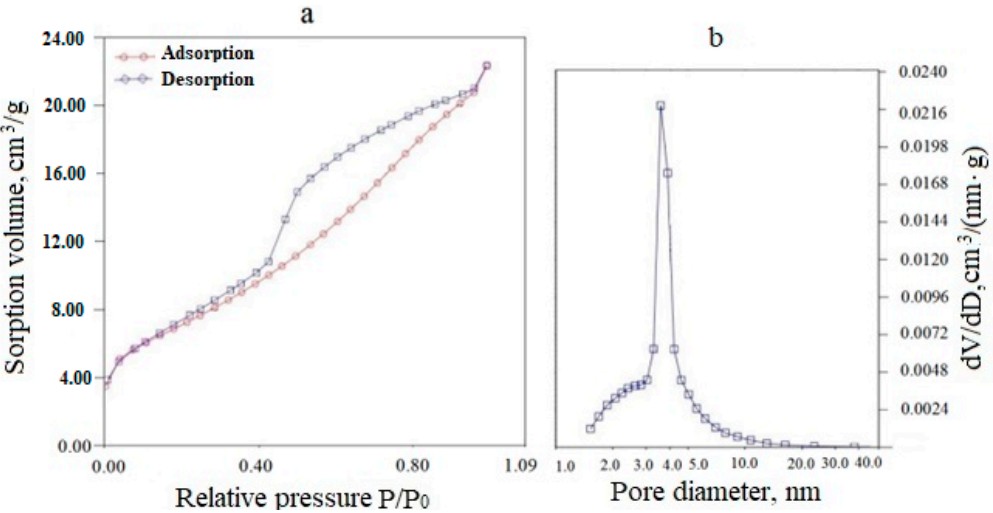

**Figure 4.** Adsorption-desorption isotherm (**a**) and differential pore size distribution (**b**) for a $(CeO_2)_{0.95}(Sm_2O_3)_{0.05}$ powder dried at 120 °C.

Similar data for a $(CeO_2)_{0.85}(Nd_2O_3)_{0.15}$ precursor powder dried at 120 °C (Figure 5) also indicated a mesoporous structure (type IV according to IUPAC classification), but the capillary-condensation hysteresis related to the H3 type corresponding to the predominance of lamellar particles forming slit-like pores.

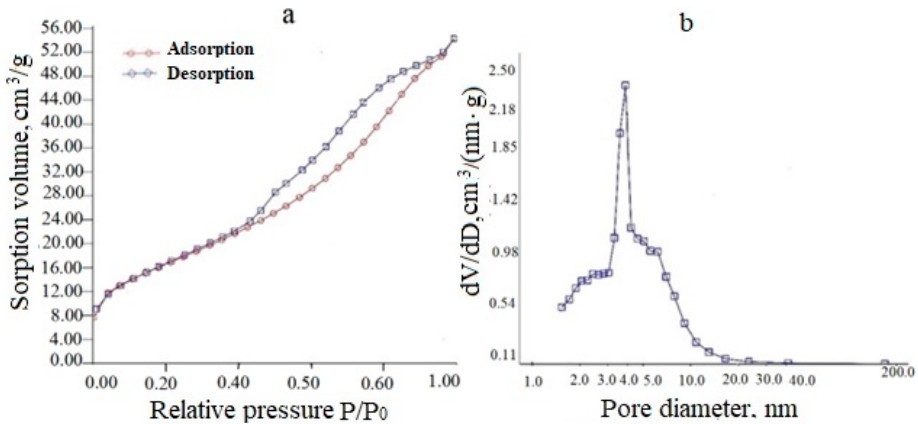

**Figure 5.** Adsorption-desorption isotherm (**a**) and differential pore size distribution (**b**) for the $(CeO_2)_{0.85}(Nd_2O_3)_{0.15}$ powder dried at 120 °C.

The textural parameters of co-precipitated $CeO_2$–$Sm_2O_3$ and $CeO_2$–$Nd_2O_3$ samples are comparatively summarized in Table 1.

**Table 1.** Textural parameters of the synthesized powders determined by BET method.

| Composition | Specific Surface Area $S_s$, $m^2/g$ | Average Pore Diameter $D_{por}$, nm | Specific Pore Volume $V_{por}$, $cm^3/g$ |
|---|---|---|---|
| Co-precipitation | | | |
| $(CeO_2)_{0.95}(Sm_2O_3)_{0.05}$ | 50 | 3.6 | 0.080 |
| $(CeO_2)_{0.90}(Sm_2O_3)_{0.10}$ | 78 | 2.5 | 0.086 |
| $(CeO_2)_{0.80}(Sm_2O_3)_{0.20}$ | 83 | 1.5 | 0.092 |
| Co-precipitation (after drying at 120 °C) | | | |
| $(CeO_2)_{0.95}(Nd_2O_3)_{0.05}$ | 119 | 3.5 | 0.111 |
| $(CeO_2)_{0.90}(Nd_2O_3)_{0.10}$ | 57 | 3.6 | 0.048 |
| $(CeO_2)_{0.85}(Nd_2O_3)_{0.15}$ | 70 | 3.7 | 0.086 |
| $(CeO_2)_{0.80}(Nd_2O_3)_{0.20}$ | 27 | 3.8 | 0.076 |
| $(CeO_2)_{0.75}(Nd_2O_3)_{0.25}$ | 41 | 3.7 | 0.093 |

The synthesized powders had a mesoporous structure with a pore size in the range of 1.5–3.8 nm, a total pore volume in the range of 0.048–0.111 $cm^3/g$, and a specific surface area of 41–119 $m^2/g$.

### 3.3. Density and Open Porosity of $CeO_2$–$Nd_2O_3$ Ceramics Synthesized Using Different Sintering Additives

To provide the required functional performances, the resulting electrolyte materials must possess an optimal density and low porosity since the target electrolytes must be gas-tight. Since $CeO_2$–$Nd_2O_3$ was found to possess an undesirably high porosity, the addition of silica ($SiO_2$) and zinc oxide (ZnO) as sintering additives before the ceramics were consolidated was attempted in order to improve this parameter. The comparative data summarized in Table 2 revealed that the addition of silica resulted in an increase in the porosity and a reduction in the material density, while in the case of ZnO application, significant growth of density and a decrease in the porosity was observed. Consequently, the following studies were performed only using zinc oxide as a sintering additive.

**Table 2.** Effect of sintering additives on the density and open porosity P of $(CeO_2)_{1-x}(Nd_2O_3)_x$ (x = 0.05, 0.10, 0.15, 0.20 and 0.25) ceramic samples.

| Composition | No Sintering Additives | | 3% $SiO_2$ | | 3% ZnO | |
|---|---|---|---|---|---|---|
| | Apparent Density $\varrho_{app}$, $g/cm^3$ | P, % | Apparent Density $\varrho_{app}$, $g/cm^3$ | P, % | Apparent Density $\varrho_{app}$, $g/cm^3$ | P, % |
| $(CeO_2)_{0.95}(Nd_2O_3)_{0.05}$ | 5.47 | 23.9 | 5.57 | 23.5 | 6.41 | 0.6 |
| $(CeO_2)_{0.90}(Nd_2O_3)_{0.10}$ | 4.67 | 29.9 | 4.30 | 32.9 | 7.02 | 3.4 |
| $(CeO_2)_{0.85}(Nd_2O_3)_{0.15}$ | 4.74 | 29.4 | 4.53 | 23.8 | 6.62 | 6.5 |
| $(CeO_2)_{0.80}(Nd_2O_3)_{0.20}$ | 6.16 | 16.2 | 4.13 | 29.6 | 6.54 | 0.6 |
| $(CeO_2)_{0.75}(Nd_2O_3)_{0.25}$ | 5.11 | 21.8 | 4.69 | 25.6 | 6.52 | 1.0 |

### 3.4. Crystal Structure Characterization of Solid Solutions in the Systems $(CeO_2)_{1-x}(Sm_2O_3)_x$ (x = 0.05, 0.10, 0.20) and $(CeO_2)_{1-x}(Nd_2O_3)_x$ (x = 0.05, 0.10, 0.15, 0.20, 0.25)

XRD data revealed the formation of the fluorite-like cubic solid solutions in both $CeO_2$–$Sm_2O_3$ and $CeO_2$–$Nd_2O_3$ systems. The sequence of this cubic solid solution formation in the

sintered $(CeO_2)_{0.85}(Nd_2O_3)_{0.15}$ sample and $(CeO_2)_{0.80}(Sm_2O_3)_{0.20}$ samples is exemplarily shown in Figures 6 and 7, respectively.

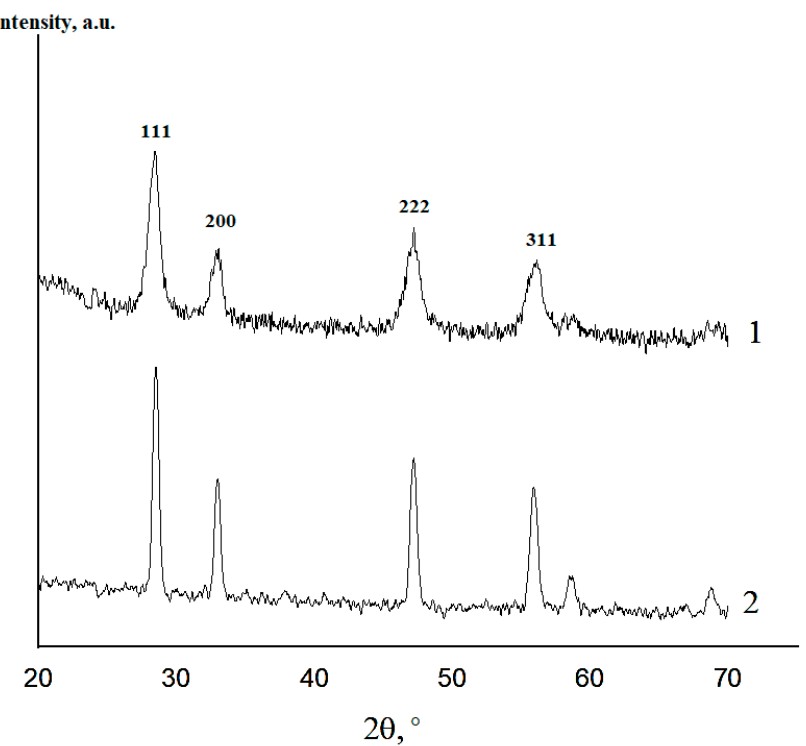

**Figure 6.** XRD profiles of $(CeO_2)_{0.85}(Nd_2O_3)_{0.15}$ powder fired at 600 °C (1) and ceramics annealed at 1300 °C (2).

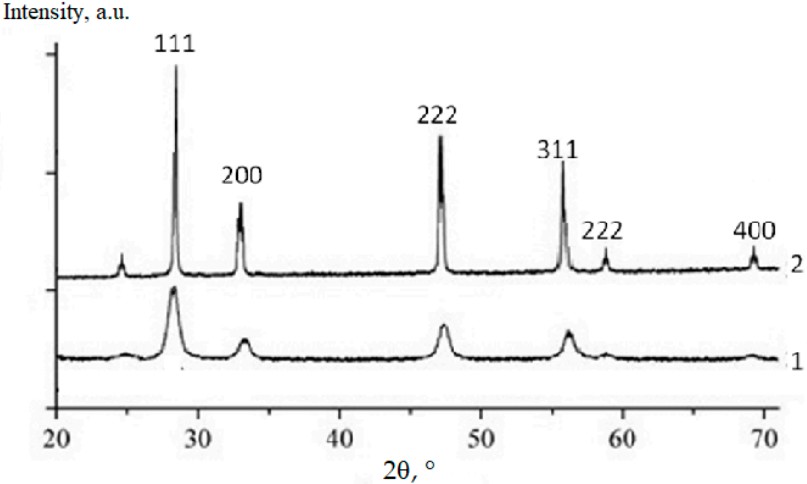

**Figure 7.** XRD profiles of $(CeO_2)_{0.80}(Sm_2O_3)_{0.20}$ powder fired at 600 °C (1) and ceramics annealed at 1300 °C (2).

The XRD spectra for all the studied compositions in the systems $(CeO_2)_{1-x}(Nd_2O_3)x$ and $(CeO_2)_{1-x}(Sm_2O_3)_x$ are presented in Figures 8 and 9, accordingly.

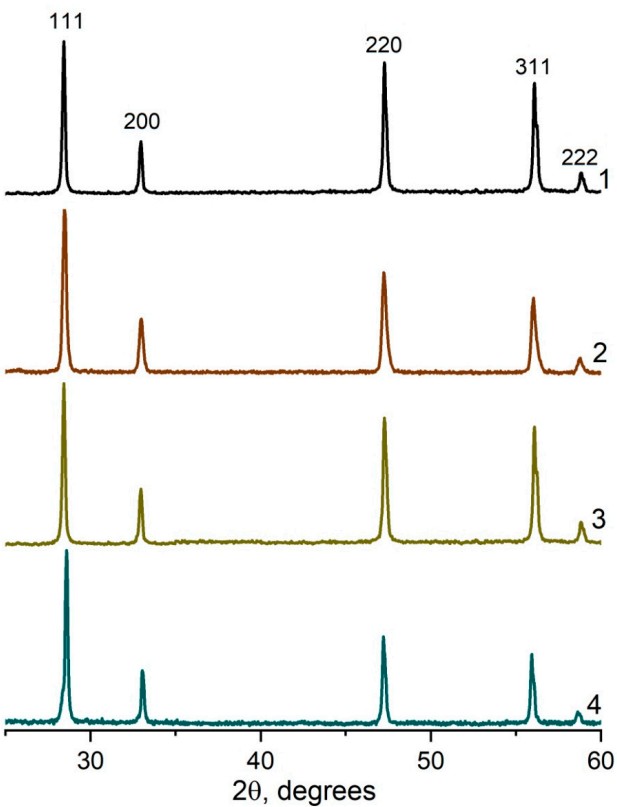

**Figure 8.** XRD profiles for $(CeO_2)_{1-x}(Nd_2O_3)_x$ ceramic materials synthesized by co-precipitation followed by annealing at 1300 °C. x = 0.05 (1), 0.10 (2); 0.20 (3), 0.25 (4).

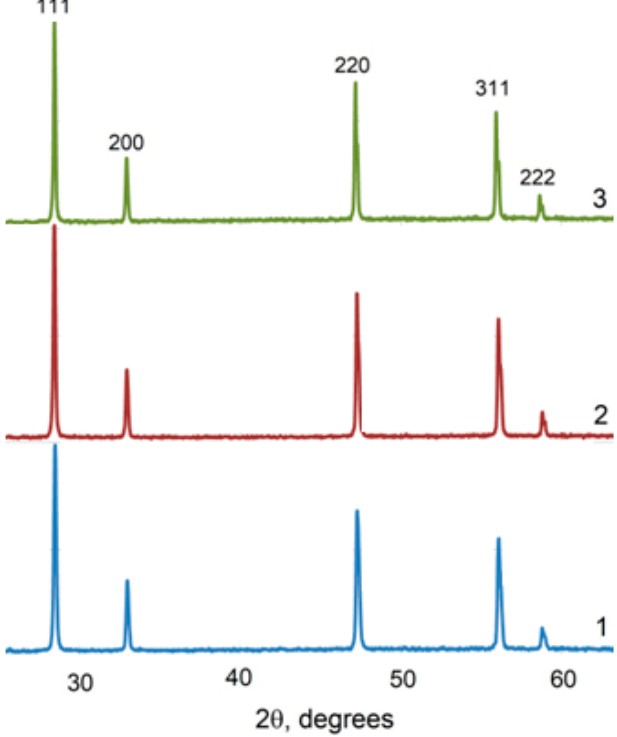

**Figure 9.** XRD profiles for $(CeO_2)_{1-x}(Sm_2O_3)_x$ ceramic materials synthesized by co-precipitation followed by annealing at 1300 °C. x = 0.05 (1), 0.10 (2); 0.20 (3).

According to the presented XRD data, the firing of the $CeO_2$–$Nd_2O_3$ sample at 600 °C for 1 h yielded finely dispersed solid solutions featuring a fluorite-type cubic structure

with a unit cell parameter a = 5.4360 Å and average CSR~14 nm, while sintering at 1300 °C resulted in a change in these parameters to a = 5.4545 Å and CSR = 88 nm. Similar values for the $CeO_2$–$Sm_2O_3$ sample at 600 °C were CSR 8 nm, at 1300 °C a = 5.4651 Å and CSR 69 nm.

Thus, the prepared ceramic samples in both systems maintained a single-phase nature in the temperature range of 600–1300 °C.

The SEM image of the $(CeO_2)_{0.90}(Nd_2O_3)_{0.10}$ sample fired at 600 °C (Figure 10a) indicated the formation of a crystalline phase including 5–25 μm sized crystals and certain pore space. The subsequent annealing at 1300 °C resulted in a complete crystallization with the formation of crystals and separate grains (Figure 10b). The granular composition of this sample is illustrated in Figure 10c, indicating the presence of 200–700 nm-sized grains with well-defined boundaries and pores between some grains.

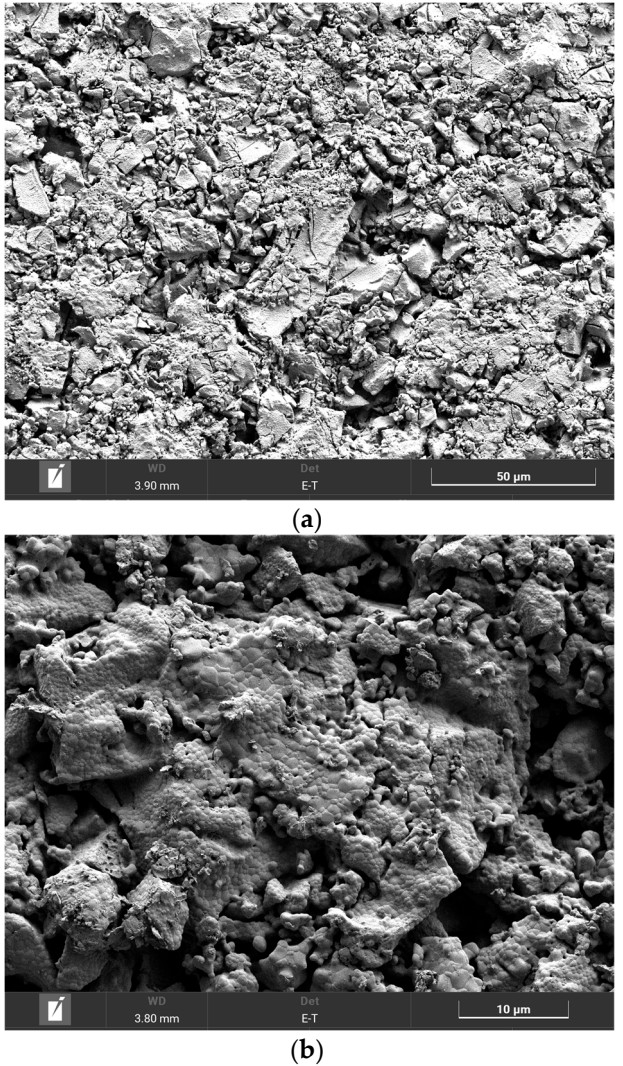

(a)

(b)

**Figure 10.** *Cont.*

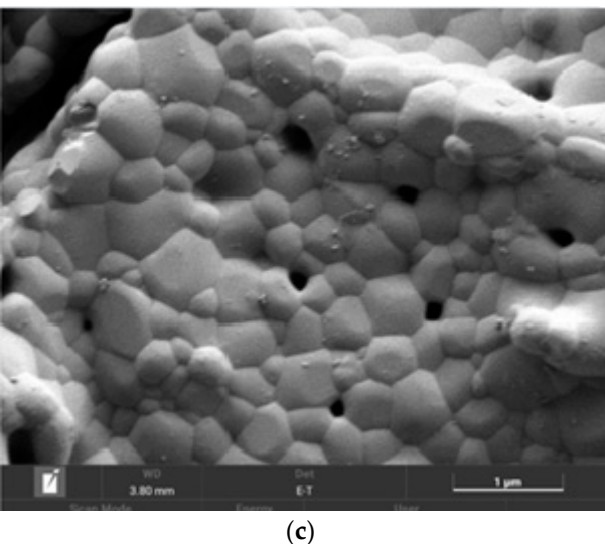

(**c**)

**Figure 10.** SEM image of the $(CeO_2)_{0.90}(Nd_2O_3)_{0.10}$ sample fired at 600 °C (**a**) and annealed at 1300 °C (**b**), as well as the granular composition (**c**).

The comparison of physicochemical properties of $CeO_2–Sm_2O_3$ and $CeO_2–Nd_2O_3$ is shown in Table 3.

**Table 3.** Physicochemical properties of $(CeO_2)_{1-x}(Sm_2O_3)_x$ (x = 0.05, 0.10, 0.20) and $(CeO_2)_{1-x}(Nd_2O_3)_x$ (x = 0.05, 0.10, 0.15, 0.20, 0.25 samples.

| Composition | Theoretical Density $\varrho_{teor}$, g/cm$^3$ | Apparent Density $\varrho_{app}$, g/cm$^3$ | Relative Density $\varrho_{rel}$, % | CSR, nm (1300 °C) | Open Porosity P, % |
|---|---|---|---|---|---|
| \multicolumn{6}{c}{$CeO_2–Sm_2O_3$} |
| $(CeO_2)_{0.95}(Sm_2O_3)_{0.05}$ | 7.23 | 6.55 | 91 | 69 | 2.0 |
| $(CeO_2)_{0.90}(Sm_2O_3)_{0.10}$ | 6.98 | 6.33 | 91 | 68 | 3.8 |
| $(CeO_2)_{0.80}(Sm_2O_3)_{0.20}$ | 6.90 | 6.25 | 91 | 65 | 6.2 |
| \multicolumn{6}{c}{$CeO_2–Nd_2O_3$ (with 3% ZnO sintering additive)} |
| $(CeO_2)_{0.95}(Nd_2O_3)_{0.05}$ | 6.82 | 6.41 | 94 | 88 | 0.6 |
| $(CeO_2)_{0.90}(Nd_2O_3)_{0.10}$ | 7.57 | 7.02 | 93 | 75 | 3.4 |
| $(CeO_2)_{0.85}(Nd_2O_3)_{0.15}$ | 7.15 | 6.62 | 93 | 73 | 1.5 |
| $(CeO_2)_{0.80}(Nd_2O_3)_{0.20}$ | 6.98 | 6.54 | 94 | 69 | 0.5 |
| $(CeO_2)_{0.75}(Nd_2O_3)_{0.25}$ | 6.96 | 6.52 | 94 | 66 | 1.0 |

*3.5. Characterization of the Surface Acid-Base Properties by Dynamic pH-Metry of Aqueous Suspensions*

The pH kinetic plots for aqueous suspensions of $CeO_2–Nd_2O_3$ and $CeO_2–Sm_2O_3$ suspensions are shown in Figure 11. These data suggest that the immersion of non-doped $CeO_2$ in water resulted only in a distinct (by 0.02–0.03) decrease in the pH of the slurry followed by growth to the initial level in 4–5 min, suggesting the passive state of the surface with a low content of active centers.

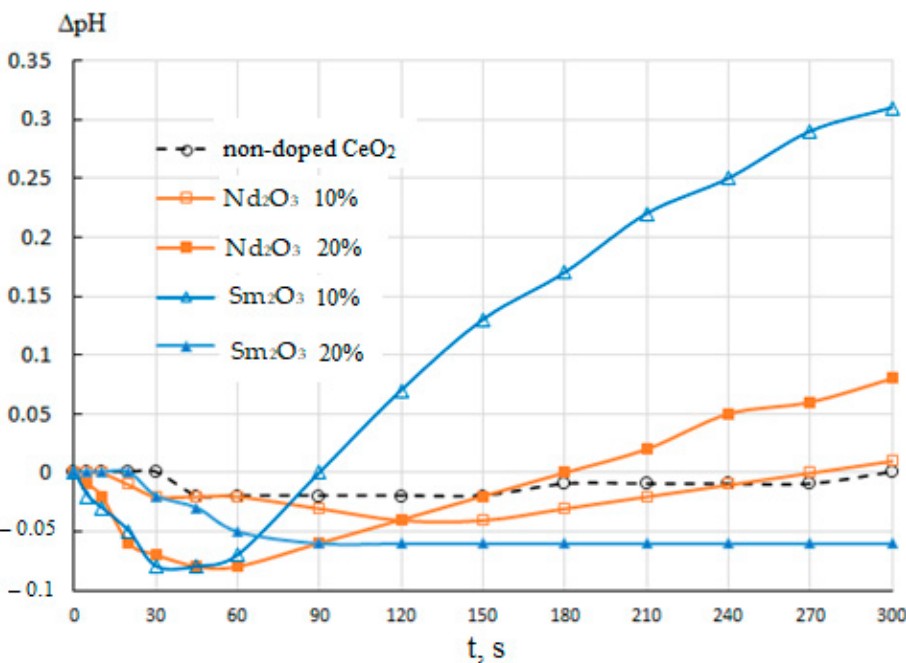

**Figure 11.** pH kinetics in aqueous suspensions of CeO$_2$–Nd$_2$O$_3$ and CeO$_2$–Sm$_2$O$_3$ samples of different compositions in comparison with non-doped CeO$_2$.

The addition of 10% Nd$_2$O$_3$ led to a slight change in the pH kinetics featuring a more significant (by 0.04) pH drop in the first 2–2.5 min, followed by a similar growth to the initial level. The increase in the Nd$_2$O$_3$ content to 20% resulted in a qualitatively similar but much more prominent effect, including a decrease of 0.08 in the first minute followed by a gradual growth to the value exceeding the initial level by 0.08. Additionally, a similar but even more pronounced effect with a decrease in pH by 0.08 after a minute, followed by an increase to a value exceeding the initial value by more than 0.3 after 5 min was observed in the case of introducing Sm$_2$O$_3$ at the amount of 10%. On the contrary, for a powder suspension containing 20% Sm, a decrease in the pH by 0.06 was observed during the first 1.5 min, followed by stabilization at this level. It should be noted that this sample differed from the rest by a pronounced hydrophobization of the surface, i.e., the absence of wetting, with the material remaining on the surface of the water, and did not sink throughout the experiment with stirring the suspension.

The observed trends in pH kinetics could be determined by the following factors. The decrease in pH in the first minutes after the immersion of the samples was probably due to the presence of Lewis acid sites (metal cations) and Broensted acid sites (OH-groups of the acid type yielding protons in an aqueous medium) on the surface. The subsequent gradual pH growth could be attributed to Broensted basic centers, i.e., hydroxyl groups dissociating more slowly with the release of the entire OH group. Broensted acidic and basic centers could be formed, respectively, by M-OH and M(OH)$_2$ groups (M = Ce, Nd, Sm). The introduction of additive atoms (Nd, Sm) apparently led to the disordering of element–oxygen bonds in the surface layer resulting in the formation of various Lewis and Broensted sites in the amounts growing with the additive content.

A drastic pH growth for the suspension of the sample containing 10% Sm$_2$O$_3$ could be accounted for by the more basic nature of this additive since, unlike neodymium, samarium can take the oxidation state +2 intrinsic to basic compounds, including the corresponding hydroxyls. The increase in the Sm$_2$O$_3$ content to 20% could lead to further growth in the content of hydroxyl groups and their condensation due to their proximity to each other, resulting in a relatively passive and hydrophobic surface covered with the element–oxygen bridging bonds with the predominance of cations (Lewis acid sites) on the surface providing the observed decrease in pH during the first minutes.

*3.6. Electrophysical Properties of $(CeO_2)_{1-x}$ $(Sm_2O_3)_x$ (x = 0.05, 0.10, 0.20) and $(CeO_2)_{1-x}(Nd_2O_3)_x$ (x = 0.05, 0.10, 0.15, 0.20, 0.25)*

The electrical conductivity of the studied samples was measured using a two-contact method at a direct current. The mechanism of electrical conductivity in these systems was mixed, including the electronic and ionic components, since the formation of oxygen vacancies was accomplished via the introduction of samarium and neodymium oxides with the lower oxidation state of the metal compared to cerium. At temperatures above 600 °C, cerium dioxide was easily reduced to $Ce^{3+}$, providing the electronic component of conductivity. Electrons also took part in charge transfer and determined the mixed nature of the conductivity of a cubic solid solution based on cerium oxide, in which the charge transport was simultaneously carried out by several types of carriers. In this regard, $CeO_2$-based materials could be considered as solid electrolytes (operating temperature range 400–600 °C) for medium-temperature solid oxide fuel cells.

The temperature dependences of the specific electrical conductivity for the prepared $CeO_2$–$Sm_2O_3$ and $CeO_2$–$Nd_2O_3$ (with ZnO sintering additive) ceramic samples are shown in Figures 12 and 13, respectively. The conductivity values at 700 °C and corresponding activation energies are summarized in Table 4.

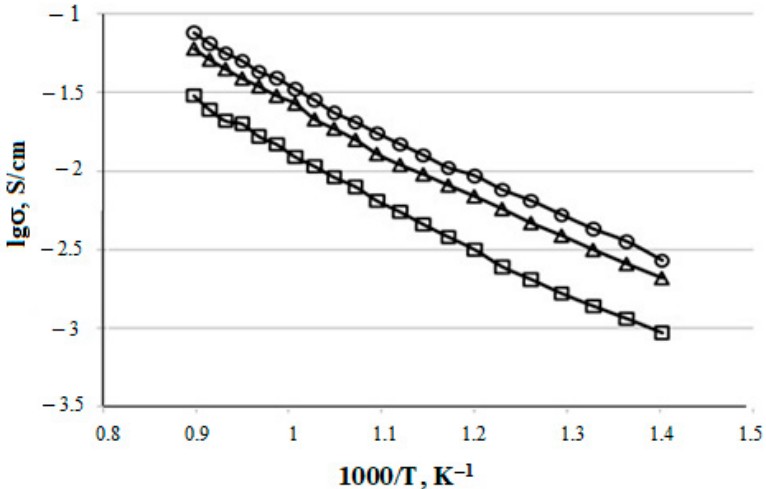

**Figure 12.** Temperature dependences of $(CeO_2)_{0.95}(Sm_2O_3)_{0.05}$ (□), $(CeO_2)_{0.90}(Sm_2O_3)_{0.10}$ (Δ) and $(CeO_2)_{0.80}(Sm_2O_3)_{0.20}$ (○) samples electrical conductivity.

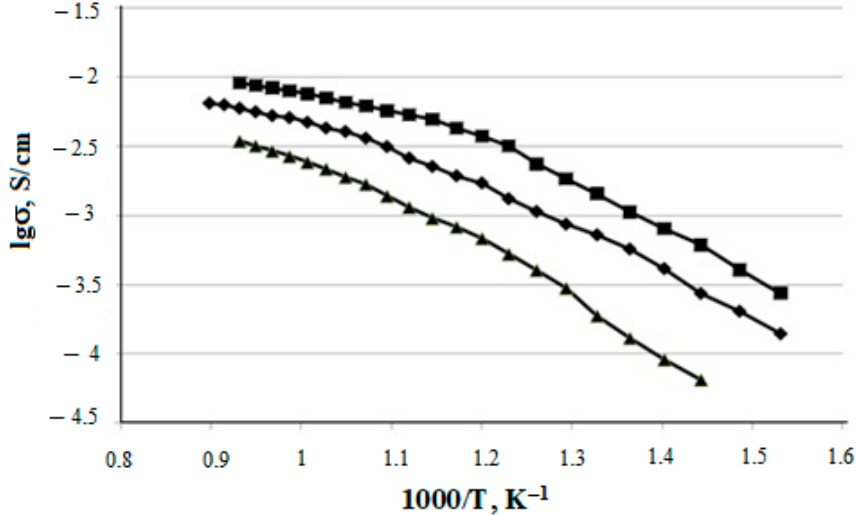

**Figure 13.** Temperature dependences of $(CeO_2)_{0.95}(Nd_2O_3)_{0.05}$ (▲); $(CeO_2)_{0.90}(Nd_2O_3)_{0.10}$ (♦); $(CeO_2)_{0.85}(Nd_2O_3)_{0.15}$ (■) samples electrical conductivity.

As shown in Figures 8 and 9, the specific conductivity of all the samples grew with the temperature in the range 500–1000 °C. Generally, the addition of $Sm_2O_3$ provided a significantly higher conductivity compared to $Nd_2O_3$. Furthermore, according to Table 4, the conductivity of $CeO_2$–$Sm_2O_3$ steadily grew with the $Sm_2O_3$ content while. in the case of $CeO_2$–$Nd_2O_3$ samples, it passed through a prominent maximum at 15 mol.% $Nd_2O_3$ and drastically dropped at higher contents of neodymium oxide. This observed behavior could probably be determined by the formation of "quasi-chemical complexes" $(Nd'_{Ce}–V_O^{\bullet\bullet})^\bullet$ involving $Nd^{3+}$ ions and mobile oxygen vacancies $V_O^{\bullet\bullet}$ [8], consequently reducing the number of oxygen vacancies.

**Table 4.** Specific electrical conductivity at 700 °C and activation energy of $(CeO_2)_{1-x}(Sm_2O_3)_x$ (x = 0.05, 0.10, 0.20) and $(CeO_2)_{1-x}(Nd_2O_3)_x$ (x = 0.05, 0.10, 0.15, 0.20, 0.25 samples.

| Composition | Specific Conductivity $\sigma \cdot 10^{-2}$, S·cm$^{-1}$ (700 °C) | Activation Energy $E_a$, eV |
|---|---|---|
| \multicolumn CeO₂–Sm₂O₃ | | |
| $(CeO_2)_{0.95}(Sm_2O_3)_{0.05}$ | 1.2 | 1.35 |
| $(CeO_2)_{0.90}(Sm_2O_3)_{0.10}$ | 2.7 | 1.31 |
| $(CeO_2)_{0.80}(Sm_2O_3)_{0.20}$ | 3.3 | 1.29 |
| \multicolumn CeO₂–Nd₂O₃ (with 3% ZnO sintering additive) | | |
| $(CeO_2)_{0.95}(Nd_2O_3)_{0.05}$ | 0.14 | 1.30 |
| $(CeO_2)_{0.90}(Nd_2O_3)_{0.10}$ | 0.36 | 1.07 |
| $(CeO_2)_{0.85}(Nd_2O_3)_{0.15}$ | 0.48 | 1.05 |
| $(CeO_2)_{0.80}(Nd_2O_3)_{0.20}$ | 0.18 | 1.01 |
| $(CeO_2)_{0.75}(Nd_2O_3)_{0.25}$ | 0.22 | 0.98 |

The highest conductivity $\sigma_{700°C} = 3.3 \times 10^{-2}$ Cm/cm in the studied temperature range 500–1000 °C was observed for the sample containing 20 mol.% $Sm_2O_3$.

The transfer numbers of ions and electrons determined by the West–Tallan method are exemplarily shown in Table 5 for the $(CeO_2)_{0.95}(Sm_2O_3)_{0.05}$ sample and in Table 6 for $(CeO_2)_{0.95}(Nd_2O_3)_{0.05}$ and $(CeO_2)_{0.90}(Nd_2O_3)_{0.10}$ systems.

**Table 5.** Transfer numbers of ions $t_i$ and electrons $t_e$ characterizing the mixed electrical conductivity of $(CeO_2)_{0.95}(Sm_2O_3)_{0.05}$ sample at different temperatures.

| T, °C | $t_i$ | $t_e$ |
|---|---|---|
| 300 | 0.85 | 0.15 |
| 400 | 0.80 | 0.20 |
| 500 | 0.78 | 0.22 |
| 600 | 0.75 | 0.25 |
| 700 | 0.73 | 0.27 |

**Table 6.** Transfer numbers of ions $t_i$ and electrons $t_e$ characterizing the mixed electrical conductivity of $(CeO_2)_{0.95}(Nd_2O_3)_{0.05}$ and $(CeO_2)_{0.90}(Nd_2O_3)_{0.10}$ samples as a function of temperature.

| Composition | T, °C | $t_i$ | $t_e$ |
|---|---|---|---|
| $(CeO_2)_{0.95}(Nd_2O_3)_{0.05}$ | 400 | 0.85 | 0.15 |
| | 700 | 0.83 | 0.18 |
| $(CeO_2)_{0.90}(Nd_2O_3)_{0.10}$ | 400 | 0.93 | 0.07 |
| | 700 | 0.85 | 0.15 |

The ion transfer number $t_i$ generally ranged from 0.73 to 0.93 and dropped with increasing temperature. The $CeO$-$Nd_2O_3$ system featured a higher part of ionic conductivity compared with $CeO$-$Sm_2O_3$, especially $(CeO_2)_{0.90}(Nd_2O_3)_{0.10}$, which provided $t_i = 0.93$ at 400 °C.

The obtained results indicated that the hydroxide coprecipitation method provided finely dispersed powders and dense low-porous nanoceramics in both studied systems with specific electrical conductivity in the range $(0.48$–$3.3) \times 10^{-2}$ Cm/cm.

## 4. Conclusions

A series of finely dispersed $(CeO_2)_{1-x}(Sm_2O_3)_x$ (x = 0.05, 0.10 and 0.20) and $(CeO_2)_{1-x}(Nd_2O_3)_x$ (x = 0.05, 0.10, 0.15, 0.20, 0.25) nanopowders with CSR~8–14 nm were prepared by the co-precipitation of hydroxides followed by freezing which was consolidated into ceramics with a fluorite-type cubic structure. The obtained ceramic materials were characterized by CSR 69–88 nm (1300 °C), open porosity in the range of 0.6–6.2%, and an apparent density of 7.02–6.25%.

Comparative studies of the porosity and density performed for ceramic samples in the systems $CeO_2$-$Sm_2O_3$ and $CeO_2$–$Nd_2O_3$ revealed that in the $CeO_2$–$Nd_2O_3$ system, the optimal values of density and open porosity could be achieved by the introduction of ZnO as a sintering additive. Samples of approximately the same density were obtained in both systems; however, $CeO_2$–$Nd_2O_3$-based ceramics featured the lowest open porosity.

For both series of ceramics samples, electrical conductivity occurred according to the vacancy mechanism. Ionic conductivity prevailed with ion transfer numbers $t_i = (0.93$–$0.73)$ in the medium temperature range (300–700) °C. The highest specific conductivity at 700 °C for $CeO_2$–$Sm_2O_3$ and $CeO_2$–$Nd_2O_3$ systems were achieved in the case of $(CeO_2)_{0.80}(Sm_2O_3)_{0.20}$ ($\sigma_{700°C} = 3.3 \times 10^{-2}$ S/cm) and $(CeO_2)_{0.85}(Nd_2O_3)_{0.15}$ ($\sigma_{700°C} = 0.47 \times 10^{-2}$ S/cm). The significantly higher conductivity for $CeO_2$-$Sm_2O_3$ samples are probably determined by their finer dispersion.

The characterization of surface acid-base properties for the obtained materials revealed that the addition of both $Sm_2O_3$ and $Nd_2O_3$ resulted in either the activation of the surface due to the distortion of element–oxygen bridging bonds or to passivation at a high (20 mol.%) $Sm_2O_3$ content, probably as a result of the condensation of the neighboring hydroxyl group upon the achievement of their critical concentration.

The conductivity of $CeO_2$–$Nd_2O_3$-based ceramics passed through a maximum at 15 mol.% $Nd_2O_3$ and drastically dropped at higher contents of neodymium oxide, probably as a result of the formation of "quasi-chemical complexes" $(Nd'_{Ce}$–$V_O^{\bullet\bullet})^{\bullet}$ involving $Nd^{3+}$ ions and mobile oxygen vacancies $V_O^{\bullet\bullet}$, consequently reducing the number of oxygen vacancies.

According to the achieved mechanical (density, open porosity) and electrophysical (the type, value and mechanism of electrical conductivity) the obtained ceramic electrolyte materials are promising as components of medium-temperature solid oxide fuel cells.

**Author Contributions:** Conceptualization, M.V.K.; methodology, M.V.K. and O.A.S.; validation, M.V.K., O.A.S. and S.V.M.; formal analysis, M.V.K., O.A.S. and S.V.M.; investigation, M.V.K., D.A.D. and S.V.M.; resources, M.V.K. and I.Y.K.; data curation, M.V.K., O.A.S., S.V.M. and D.A.D.; writing—original draft preparation, M.V.K.; writing—review and editing, M.V.K. and S.V.M.; visualization, D.A.D. and S.V.M.; supervision, M.V.K., I.Y.K. and O.A.S.; project administration, I.Y.K. and M.V.K.; funding acquisition, I.Y.K. All authors have read and agreed to the published version of the manuscript.

**Funding:** The study is supported by the State Assignment for the Institute of Silicate Chemistry of the Russian Academy of Sciences (State registration number 0081-2022-0007).

**Institutional Review Board Statement:** Not applicable.

**Informed Consent Statement:** Not applicable.

**Data Availability Statement:** Not applicable.

**Acknowledgments:** The authors are thankful to Anastasia Kovalenko (Institute of Silicate Chemistry of the Russian Academy of Sciences) for their assistance with the dynamic pH-metry characterization of the studied materials.

**Conflicts of Interest:** The authors declare no conflict of interest.

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
