# Peer review of "Comparative Study of Physicochemical Properties of Finely Dispersed Powders and Ceramics in the Systems CeO2–Sm2O3 and CeO2–Nd2O3 as Electrolyte Materials for Medium Temperature Fuel Cells"

_ceramics, doi:10.3390/ceramics6020073_

Round 1

Reviewer 1 Report

In this paper, the authors studied the physicochemical properties of finely dispersed powders and ceramics in the CeO2–Sm2O3 and CeO2–Nd2O3 systems. The compositions that forms highest ionic conductivity are identified and the general properties (density, open porosity, type and mechanism of electrical conductivity) of the obtained ceramic materials have been characterized. I only have a few questions for the improvement of the manuscript. 

1.     While lots of materials properties (density, open porosity, type and mechanism of electrical conductivity) are characterized in this paper, how such properties individually and collectively contribute to the solid oxide electrolytes of better performance for fuel cells are not clear.

2.    The authors argue that the electrical conductivity occurs according to the vacancy mechanism. Can they give more evidence? 

3.    There are some typos, like that on page 3, line 96; page 12, line 364.

Reviewer 2 Report

The authors synthesized a series of finely dispersed (СeO2)1-x(Sm2O3)x and (СeO2)1-x(Nd2O3)x powders based on co-precipitation of the hydroxides by liquid-phase method followed by low-temperature treatment. The influence of elemental and concentration composition on the microstructure and the physicochemical properties of the powders and ceramics were also studied. However, the following issues need to be carefully addressed.

1. Can the freezing process remove the crystallized water from the gel? How does the freezing process reduce the crystallization temperature of СeO2-Sm2O3/СeO2-Nd2O3 solid solution? Does the reduction of crystallization temperature have any effect on the microstructure of the powder?

2. When the content of SiO2 in the system is more than 0.005% (ω), the grain boundary resistance mainly comes from the formation of "blocked silicate film" by SiO2 impurities, which hinders the migration of grain boundary oxygen ions and reduces the grain boundary conductivity. Please explain the reasons for choosing SiO2 as a sintering aid for СeO2-Nd2O3.

3. Table 3. The density of electrolytes prepared from СeO2-Sm2O3/СeO2-Nd2O3 powder is less than 95% at the sintering temperature of 1300 ℃. This density seems relatively low for the electrolyte of SOFC.

4. Compared with СeO2-Sm2O3, СeO2-Nd2O3 has a higher ionic conductivity. Why the former has a lower conductivity? What factors cause the lower conductivity?

5. The SEM images of the powder can be provided to better illustrate the dispersibility of the powder.

Reviewer 3 Report

The authors have reported a similar proposal elsewhere, but this version contains more details and results by inducing other precursors in the experimentation and discussion sections.

But, the problem of weak discussion of structural and morphological properties still exists. Also the special issue related to this publication is “Composite Nanopowders: Synthesis and Applications”. However, the authors did not address the study of the compounds as composite materials. For this, a thorough study of the structural and morphological parts must be given (see the following articles:

1- https://doi.org/10.1016/j.ceramint.2020.05.038

2- https://doi.org/10.3390/ceramics4030035

3- https://doi.org/10.1016/j.materresbull.2017.09.064

In addition, authors should cite some articles published by this Journal of ceramics or in other mdpi journals, here are some suggestions:

https://doi.org/10.3390/ceramics4030035

https://doi.org/10.3390/nano11092231

https://doi.org/10.3390/molecules21050644

Round 2

Reviewer 2 Report

The authors have tried to answer my concerns.

Author Response

In addition to the previous revision, more details about the structure of the synthesized materials are presented, including XRD spectra for all the sintered ceramics samples (Fig. 8, 9) and SEM image illustrating their granular composition (Fig. 10c). 

Reviewer 3 Report

The answers of the authors are not sufficient to respond to my comments. Fro example the SEM images are not made in the correct way to give information on the morphology of the samples: the size of the grain boundaries, the contrast of the compositions, the pores... 

For the XRD diffractograms: the authors said that the studied compounds are composite materials, for that, they selected this Special Issue. Indeed, they report XRD patterns of solid solution compounds. For this reason, I asked to plot each XRD diffractogram and indexed all peaks. 

Author Response

An additional SEM image illustrating the granular composition and morphology of the prepared ceramic materials with well-defined grain boundaries and pores is presented in Fig. 10c with the corresponding discussion is highlighted in yellow.

Additional XRD spectra for all the prepared ceramic materials are shown and highlighted in Figures 8 and 9 with the description of al the observed peaks.

The considered powdered materials are essentially two-component systems which under certain conditions (changes in the ratio between the components, increase of the sintering additive amount, as well as the application of different processes for the sintering additive introduction) can become composites containing inclusions of various compounds in addition to the cubic solid solution.  These issues will be addressed in our subsequent studies.